# SPMC: Self-Purifying Federated Backdoor Defense via Margin Contribution

Wenwen He [* 1 2]   Wenke Huang [* 1]   Bin Yang [1]   Shukan Liu [3]   Mang Ye [1]

## Abstract

Federated Learning (FL) enables collaborative training with privacy preservation but is vulnerable to backdoor attacks, where malicious clients degrade model performance on targeted inputs. These attacks exploit FL decentralized nature, while existing defenses, based on isolated behaviors and fixed rules, can be bypassed by adaptive attackers. To address these limitations, we propose **SPMC**, a marginal collaboration defense mechanism that leverages intrinsic consistency across clients to estimate inter-client marginal contributions. This allows the system to dynamically reduce the influence of clients whose behavior deviates from the collaborative norm, thus maintaining robustness even as the number of attackers changes. In addition to overcoming proxy-dependent purification's weaknesses, we introduce a self-purification process that locally adjusts suspicious gradients. By aligning them with margin-based model updates, we mitigate the effect of local poisoning. Together, these two modules significantly improve the adaptability and resilience of FL systems, both at the client and server levels. Experimental results on a variety of classification benchmarks demonstrate that SPMC achieves strong defense performance against sophisticated backdoor attacks without sacrificing accuracy on benign tasks. The code is posted at: *https://github.com/WenddHe0119/SPMC*.

## 1. Introduction

Federated Learning (FL), as a decentralized machine learning paradigm (McMahan et al., 2016; Bonawitz, 2019), en-

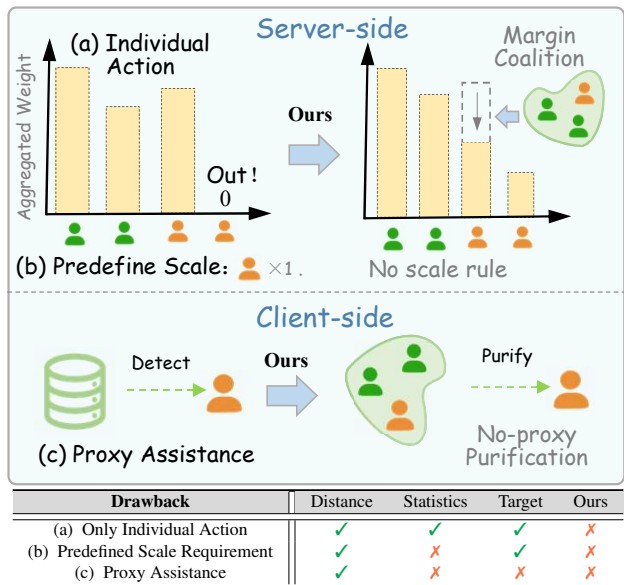

Figure 1. **Motivation**. Illustration of the limitations of existing defenses and the advantages of the proposed method from both server-side and client-side. The comparison table highlights that our method uniquely avoids reliance on three drawbacks. 👤 denotes attacker and 👤 denotes benign client.

| Drawback | Distance | Statistics | Target | Ours |
|---|---|---|---|---|
| (a) Only Individual Action | ✓ | ✓ | ✓ | ✗ |
| (b) Predefined Scale Requirement | ✓ | ✗ | ✓ | ✗ |
| (c) Proxy Assistance | ✓ | ✗ | ✗ | ✗ |

ables multiple devices to collaboratively train a global model while preserving data privacy by storing data locally on end devices. The general approaches (McMahan et al., 2017; Blanchard et al., 2017) aggregate parameters from participating devices and subsequently redistribute the global model (averaged parameters) for further training, aiming to learn a high-quality model without centralizing private data. However, this distributed learning framework complicates the verification of each participant's trustworthiness, rendering it vulnerable to backdoor attacks (Chen et al., 2017; Gu et al., 2019; Li et al., 2022; Fung et al., 2020). In such attacks, an attacker may insert triggers into one or more local models, causing the global model to exhibit specific malicious behaviors when these triggers are activated. For instance, in the context of autonomous driving, an attacker could place special stickers (e.g., smiley faces) on a stop sign, misleading the system into interpreting the stop sign as a speed limit, thereby exemplifying a backdoor attack in a real-world scenario (Gu et al., 2019). Therefore, effectively addressing backdoor attacks is essential for ensuring the reliability of federated learning in real-time applications.

*Equal contribution [1]National Engineering Research Center for Multimedia Software, School of Computer Science, Wuhan University, Wuhan, China [2]School of National Cyber Security, Wuhan University, Wuhan, China [3]School of Electronic Engineering, Naval University of Engineering, Wuhan, China. Correspondence to: Bin Yang <yangbin_cv@whu.edu.cn>, Mang Ye <yemang@whu.edu.cn>.

*Proceedings of the 42^{nd} International Conference on Machine Learning*, Vancouver, Canada. PMLR 267, 2025. Copyright 2025 by the author(s).

Existing defenses against backdoor attacks can be categorized into server-side defenses (Nguyen et al., 2024) and client-side defenses (Zhu et al., 2023b; Huang et al., 2024a). Both strategies either leverage individual distance differences (Blanchard et al., 2017; Fung et al., 2018; Shejwalkar & Houmansadr, 2021), focus on outlier resilience by the overall statistical characteristics of local updates (Guerraoui et al., 2018; Yin et al., 2018), or optimize local updates based on target thresholds (Li et al., 2019; Ozdayi et al., 2021; Xie et al., 2021). However, these approaches rely on two primary assumptions: individual behavior and passive purification. Individual action regarding client-to-whole distances overlooks the behavioral communication between clients. Passive purification, which relies on predefined scaling rules or an additional clean dataset, can easily compromise data privacy and overlook biases in poisoned participant numbers (Li et al., 2020b; Ye et al., 2025b; Fang et al., 2025).

Inspired by the Shapley value (SV) from game theory (Ghorbani et al., 2020; Jiang et al., 2023), we observe an important phenomenon in FL. The contribution made by a backdoor attacker differs significantly from that of a coalition that excludes the attacker. This difference can serve as a reliable indicator for identifying malicious clients. To quantify this, we define the inter-client margin contribution, which measures the impact of a client's participation by comparing its local model parameters with those of the corresponding coalition model. This approach addresses the limitations of existing assumptions and enhances system robustness by effectively capturing and mitigating abnormal client behavior.

Motivated by the perspective of margin contribution in game theory, we introduce a Self-purifying Backdoor Defense via Margin Contribution (**SPMC**), including client gradient optimization and server aggregation for active purification. Specifically, the server receives malicious gradient orientations markedly different from those of benign participants, resulting in a substantial margin difference. We argue that the difference in model parameters with the contribution of the margin is crucial to accurately identify attackers. Consequently, we propose margin difference aggregation, wherein, for each client, we measure the margin contribution by the cosine similarity based on the parameter differences between that client and other clients (the coalition), allowing the purification of poisoning effects without a predefined scale. We adaptively adjust the importance of each participant according to their margin contribution, assigning lower aggregation weights to distorted clients with significant margin differences while encouraging participants with smaller margin differences to contribute more to the collaborative effort. In the client-side scenario, to mitigate the impact of attackers continuously influencing the global model with toxic data, we further align the direction of the participants local model gradients with the direction of the gradient representing general knowledge (Guo et al., 2022; Zhu et al.,

2023a) to realize self-purifying. This gradient measures the discrepancy between local model predictions and the knowledge communicated by the coalition model, thereby preventing overfitting to poisoned samples (Wang et al., 2023; Rame et al., 2022; McMahan et al., 2017; Shi et al., 2021; Yang et al., 2024). Extensive experiments demonstrate that our defense method maintains high model performance and robustness simultaneously, a feat not previously achieved.

Our main contributions are as follows:

- We concentrate on the backdoor robustness in federated learning and reveal that existing approaches only focus on individual action and overly depend on the predefined scale with proxy assistance to enhance backdoor attack defenses, resulting in the global model's vulnerability against changes in the number of attackers. We address the lack of flexible purification of malicious impacts through client margin difference contributions, realizing non-proxy distillate for local model guidance.

- To establish a clear distinction between the malicious and benign participants in the global model, we introduce a self-purifying margin difference contribution defense (SPMC). In server aggregation, our method quantifies parameter differences between the local model and the margin coalition model, governing server aggregation without any predefined scale. On the client side, we realize non-proxy distillation to guide local gradients to align with the margin models' direction, preserving all local gradients in a benign direction.

- We conduct experiments on various datasets, including CIFAR-10/100, MNIST, and FashionMNIST, under backdoor attack conditions. The effectiveness of the proposed SPMC is validated, confirming the indispensability of the foundational module.

## 2. Related work

### 2.1. Overview of Federated learning

Federated Learning (Huang et al., 2024b; Wan et al., 2025) has become a popular Machine Learning framework because it allows clients to train models in a decentralized manner without having to share any private datasets. In the FL framework, data for learning tasks is acquired and processed locally at the edge node, and only the updated Machine Learning parameters are transmitted to the central orchestration server for aggregation. It consists of four phases: FL initialization, local model training, local model updating, and aggregation. If we make certain assumptions about the type of attack and limit the number of malicious clients. Aggregation techniques can produce robust training models in some cases (Huang et al., 2023b; 2022). Fedavg (McMahan et al., 2016) is widely used in FL for both attack and defense scenarios, especially in work on backdoor attacks and defences (Bagdasaryan et al., 2020; Nguyen et al., 2022;

Shen et al., 2016; Nguyen et al., 2020; Muñoz-González et al., 2019; Fung et al., 2020). While the algorithm also allows for weighting the contributions of different clients, this also makes the system more susceptible to manipulation, as malicious users can use this to increase their impact.

### 2.2. Defense Against the Backdoor Attack

Malicious backdoor attacks pose a serious threat to federal systems. Existing backdoor defense schemes against backdoor attackers can be classified into three categories:

**Distance Differential Defence** (Blanchard et al., 2017; Fung et al., 2018; Tian et al., 2021; Zheng et al., 2021; Han et al., 2023; Fereidooni et al., 2024). Mainly through the difference in distance between local updates and server aggregated updates, clients with obviously far from the overall direction are considered evil and excluded from the aggregation process. For example, Sageflow (Park et al., 2021) combines entropy filtering and loss to measure the difference between the local model and the ideal distribution weighting of the server model. FLTrust (Cao et al., 2021a) collects a clean, small training dataset and thus introduces Relu-clipped cosine similarity to allocate high trust scores for those reliable clients. Distance discrepancy defense mostly focuses on local-to-server exchanges, thus neglecting inter-local edge exchanges. In addition, the security of the agent dataset determines whether the ideal distribution is fitted to the benign update, aggregating the robust model.

**Statistical Distribution Defence** (Yin et al., 2018; Guerraoui et al., 2018; Pillutla et al., 2022; Zhang et al., 2022b; Cao et al., 2023) constructs different statistical criteria to select and exclude bad users. For example, RFA (Pillutla et al., 2022) calculates the geometric median with an alternating minimization function. FLDetector (Zhang et al., 2022b) considers the historical client updates and votes for those with large discrepancies between the predicted and received client updates as attackers. However, the above two approaches require complex hyper-parameter configurations to adapt to heterogeneous federation scenarios, striking a blow to the generalization and robustness of backdoor defense in terms of generalization and robustness.

**Objective Optimized Defence** (Wu et al., 2020; Li et al., 2019; Ozdayi et al., 2021; Cao et al., 2021b; El-Mhamdi et al., 2021; Panda et al., 2022) locally sets threshold objectives to regulate the global paradigm and weed out bad users. For example, RLR (Ozdayi et al., 2021) adjusts the learning rate considering the agent update direction and slows down the learning rate for potentially malicious participants with a large number of locally conflicting update directions. CRFL (Xie et al., 2021) sets pruning thresholds for the global model paradigm, scales the model scrub eucalyptus tree of bad users proportionally, and adds noise to reduce the malicious influence of bad users. However, the

above methods require appropriate thresholds to regulate the benign update of global paradigms.

The above defenses mainly rely on individual action, the predefined rule, and proxy assistance, as shown in Fig. 1. These rules are crucial to ensure the difference between benign and malicious updates and the correctness of the overall direction of the client's local updates, which are serious rule challenges for different realistic settings.

## 3. Preliminaries

**Threat Model**: Backdoor attacks have been extensively studied against centralized learning to manipulate the centralized model by inserting some triggers during training. Specifically, we define $\tau$ as the trigger pattern and $m$ as the trigger location mask. The modified backdoor instance is represented as $\widetilde{\xi} = (\widetilde{x}, \widetilde{y})$. For $\widetilde{x}$, we apply the formula $\widetilde{x} = (1-m) \odot x + m \odot \tau$, incorporating the trigger pattern $\tau$ into the original instance $x$ at locations specified by the mask $m$ (Liao et al., 2018). We then alter the original label $y$ to the predefined attack target $\widetilde{y}$. For any input $x$, the $k$th client dataset $D_k$ has the following output:

$$D_k = \begin{cases} \widetilde{D}(\widetilde{x}, \widetilde{y}), \exists x \in \widetilde{\xi} \\ D(x, y), \forall x \notin \widetilde{\xi} \end{cases}, \qquad (1)$$

where $\widetilde{D}$ denotes the compromised dataset that exploits the flip-flop label attack or proportionally increases the weight of the malicious model (Xie et al., 2020).

**Local Update**: In each iteration of FL, each client updates the local model based on its local dataset (Li et al., 2020a). We denote $D_k$ as the $k$th client's local dataset. Under heterogeneous federated learning, the $k$th client's model parameter distribution $I_k^t$ in the $t$th epoch:

$$I_k^t \leftarrow w_g^{t-1} - \eta \nabla \mathcal{L}(w_g^{t-1}, D_k), \qquad (2)$$

where $\eta$ denotes the learning rate and $I_k^t$ is determined by the local dataset $D_k$, global parameter $w_g^{t-1}$, and updates model parameters by the loss function $\mathcal{L}$.

**Server Aggregation**: In federated learning, the central server receives the updated model parameters from the local and updates the global model according to a certain aggregation strategy (usually federated aggregation (McMahan et al., 2017)). The global model parameters in the $t$th epoch can be expressed as: $w_g^t = \sum_{k=1}^{N} \alpha_k I_k^t$, where $\alpha_k$ denotes the weight of the $k$th client, which must satisfy $\sum_{k=1}^{N} \alpha_i = 1$.

## 4. Methodology

### 4.1. SPMC: Self-Purifying Margin Contribution

**Motivation**. Most existing defenses to backdoor attacks focus on individual actions that reward clients based on global distribution but ignore inter-client communications.

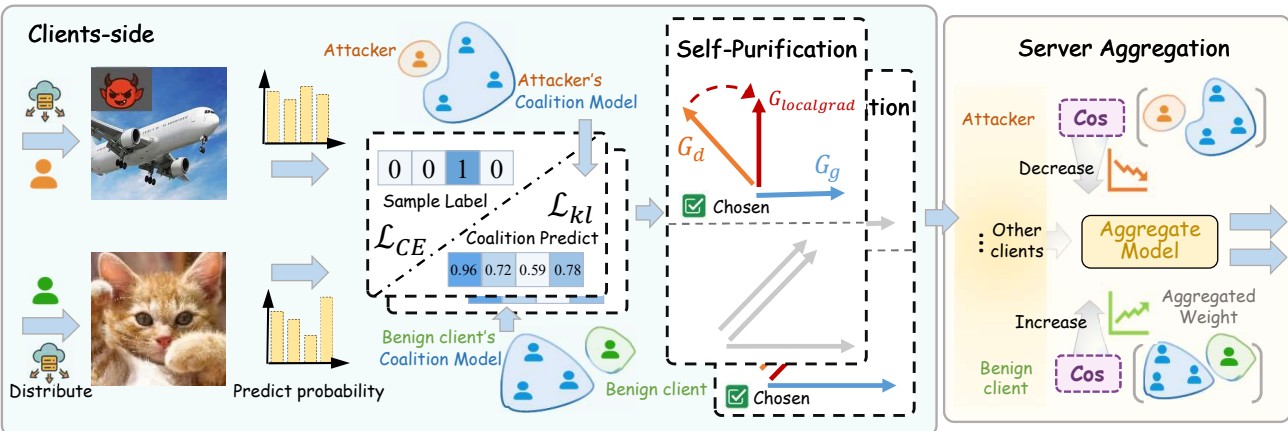

*Figure 2.* **Overview** of Self-Purifying Margin Contribution (SPMC). See Sec. 4. On the client side, we utilize the coalition aggregation model for gradient alignment, wherein the updated local model parameters are sent to the server side. On the server side, margin differences among these parameters are assigned appropriate weights to facilitate the aggregation of the global model.

To address these problems, our idea is to estimate each client's contribution to other clients, and further use the contribution to guide the direction of local updates and identify malicious attackers (Jiang et al., 2022). This idea is inspired by the Shapley Value, a classical method to quantify participants' contributions in cooperative game theory. Our research is the first to apply SV for backdoor defense.

For $N$ clients, we have $I^N = \{I_k\}_{k=1}^N$ as the set of local client distributions. According to Eq. 2, the distribution $I_k$ is supported on a space $\mathcal{D}_k = \mathcal{X} \times \mathcal{Y}$, $\mathcal{X}$ and $\mathcal{Y}$ is input and output respectively. The coalition $S \sim \mathcal{D}^M$ is a set of clients, such that $\|S\| = M$, where $M$ denotes the number of clients in the coalition. Let $\Gamma \in [0,1]$ denote the utility function, and $\Gamma(S)$ represents the value of $S$ coalition. We express $\Gamma$ as the target $\Gamma \leftarrow \min \mathbb{E}_{D_k \sim D^M}[S(\mathcal{L}(w_g, D_k))]$ for each coalition. We define SV as below:

$$\phi_k(k; \Gamma; D; N) = \mathop{\mathbb{E}}_{M \sim [N], \Gamma \sim D^{M-1}}[\Gamma(S \cup \{k\}) - \Gamma(S)]. \quad (3)$$

From this definition, the SV of a client is its expected margin contribution in $\Gamma$ to a set of client coalitions $S$. To calculate SV, we need to consider all possible client coalitions, i.e., all subsets of $N$ clients. The cost will be exponentially increased with respect to the number of clients $N$, that is, $O(2^N)$. Such computation is extremely expensive, even with a small number of clients. Therefore, it is critical to find an efficient solution for client valuation. By analyzing the SV definition, we notice that the key is to measure the value with and without a certain client with respect to all possible combinations of other clients (shown in Fig. 1). Therefore, we propose an efficient approximation, by directly measuring the contribution of client $k$ to others $(N \backslash \{k\})$. We define our new value $\phi_k$ as:

$$\phi_k(k; \Gamma; D; N) = \mathop{\mathbb{E}}_{\Gamma \sim D^N}[\Gamma(N \backslash \{k\}) - \Gamma(\{k\})]$$
$$\Gamma(\{k\}) = I_k, \quad (4)$$

where $\phi_k$ is our proposed function to measure the margin contribution of client $k$. We focus on verifying whether a new client contributes benignly to the existing client. Suppose the new client differs significantly from the existing client in terms of the target goal; in that case, we believe that the federation resulting from the new client joining the existing client will produce a malicious contribution. Backdoor defense in federated learning aims to penalize clients whose coalitions generate malicious contributions.

In the rest part of this section, we investigate how to apply margin contribution on both the server-side and client-side for attacker identification and self-purification ( Fig. 2).

### 4.2. Margin Contribution Aggregation

**Margin Aggregation**: According to Eq. 4, we first compute the coalition $\Gamma(S \backslash \{k\})$ on the server side as a standard to measure the likelihood of potential attackers. The attacker trains malicious local model parameters due to poisoned private datasets. Therefore, we secondly similarly utilize coalition $\Gamma(S \backslash \{k\})$ as a standard for updating local model parameters on the client side. Since the global model overwrites the local model in round $t$, we need to store the updated local model parameters for round $t-1$ and recalculate the $k$th coalition. We define $I_{S_k}$ as a simplistic modelling description of the coalition $\Gamma(S \backslash \{k\})$ in the $t$th epoch:

$$I_{S_k} \leftarrow \Gamma(N \backslash \{k\}) = \underbrace{\sum_i^{S_k} \alpha_i I_i^t}_{\textbf{Server}} = \underbrace{\frac{w_g^t - \alpha_k \times I_k^{t-1}}{1 - \alpha_k}}_{\textbf{Client}}, \quad (5)$$

where $S_k = (\{1, 2, \ldots, n\} \backslash \{k\})$ represents a coalition without the $k$th client and $\alpha_k$ means the $k$th client's aggregation weight reconstructed via margin contribution.

**Difference Computation**: According to Eq. 4, under Federated Learning (FL), the margin contribution of client $k$ is quantified as the discrepancy in model parameters between

client $k$ and the coalition excluding that client. We employ the cosine similarity measure to evaluate the alignment between the local client and the margin aggregation model (the coalition model) (Huang et al., 2023a), as it focuses on directional alignment rather than magnitude, making it more robust in high-dimensional, heterogeneous FL settings. We further compare cosine similarity with Euclidean and Wasserstein distances, with results reported in Table 5 in Appendix C. Consequently, the Equation can be simplified as: $\phi_k(k; \Gamma; D; N) = \frac{I_{S_k} \cdot I_{k \notin S_k}}{\|I_{S_k}\| \times \|I_k\|}$. As indicated by the formula, a high level of similarity suggests that the potential attacker is significantly distant from the parameters of the coalition model for accurately identifying that attacker.

**Reweighted Aggregation**: We note that the existing aggregation weights $\alpha_k$ are typically based on data scale: $\alpha_k = \frac{N_k}{\sum_k^K N_k}$ or participant size $\alpha_k = \frac{1}{K}$. We argue that popular aggregation methods ignore the reliability of the margin contribution and assign higher weights to potential attackers that are far from their corresponding margin coalition. Therefore, we reconfigure the aggregation weights of the clients via the margin contribution set $\phi$:

$$\phi = [\Gamma(N\backslash\{1\}) - \Gamma(\{1\}), \ldots, \Gamma(N\backslash\{n\}) - \Gamma(\{n\})]$$
$$\text{Cosine} \Downarrow \widehat{\phi}_k \in \max -\phi_k$$
$$\widehat{\phi} = [\widehat{\phi}_1, \ldots, \widehat{\phi}_k, \ldots, \widehat{\phi}_n], \quad (6)$$
$$\alpha_k = \frac{\sigma(-\widehat{\phi}_k)}{\sum_{k'} \sigma(-\widehat{\phi}_{k'})}.$$

Specifically, for local models, the impact of their model parameters is highlighted when the similarity between their model parameters and the coalition model parameters is high, and their impact is penalized when it's low.

### 4.3. Local Gradient Alignment

**Motivation**. The central server broadcasts $w_g$ to each participant in the form of $w_k \leftarrow w_g$. Participants perform the local optimal function to fit the local distribution according to their private dataset $D_k$. From Eqs. 1 and 2, benign and malicious fit different distributions and naturally hold different gradient directions (Rame et al., 2022). Specifically, malicious changes the original label $y$ to an attack target $\widetilde{y}$ by trigger pattern $\tau$, resulting in a gradient direction that deviates significantly from the benign participants' direction to influence the global model subtly:

$$\mathcal{L}_k(w_k^t, D_k) = \frac{1}{|D_k|}[\sum_{\xi \in D_k} \underbrace{\mathcal{L}_{CE}(x, y)}_{\text{Benign}} + \sum_{\widetilde{\xi} \in D_k} \underbrace{\mathcal{L}_{CE}(\widetilde{x}, \widetilde{y})}_{\text{Backdoor}}], \quad (7)$$

where $\widetilde{\xi} = (\widetilde{x}, \widetilde{y})$ denotes the modified backdoor trigger. $\mathcal{L}_{CE} \leftarrow -\sum y_i log(p(t_i|x))$ is the loss function typically used by traditional methods, optimizing the model parameter by minimizing the negative log-likelihood. But $\mathcal{L}_{CE}$ relies heavily on anti-overfitting techniques (Zhou et al.,

2022), encouraging the attacker to extend the malicious impact once again. We propose an effective self-purifying updates paradigm that aligns local malicious gradient direction with the general knowledge of margin coalition models.

**Non-proxy Distillation**. Due to the great success of knowledge distillation in catastrophic forgetting (Phuong & Lampert, 2019; Hinton et al., 2015; Kirkpatrick et al., 2017), researchers have applied knowledge distillation in backdoor attacks, expecting the model to be more discriminative to the attacker. We hope to leverage margin contributions to distill agentless datasets for self-purification. We utilize the margin coalition model predictions as general knowledge and compare the model predictions to the general knowledge to align the local gradient direction. Specifically, we obtain the domain-specific direction by calculating the cross-entropy $\mathcal{L}_{CE}(w)$ between the local model prediction $p(t_i|x)$ and the true target $y$, and the general knowledge direction based on the Kullback-Leibler (KL) divergence between $p(t_i|x)$ and the marginal model union prediction $p_{S_k}(t_i|x)$:

$$\mathcal{L}_{kl}(w) = -\sum p_{S_k}(t_i|x) \times log \frac{p(t_i|x)}{p_{S_k}(t_i|x)}. \quad (8)$$

We denote the gradients of $\mathcal{L}_{kl}(w)$ and $Lce(v)$ as $G_g = \nabla \mathcal{L}_{kl}(w)$ and $G_d = \nabla \mathcal{L}_{CE}(w)$, respectively. The relations between $G_g$ and $G_d$ are twofold. Firstly, the fact that their angle is less than 90° indicates that the local knowledge possessed by the local client does not conflict with the general knowledge, In this case, we safely set the updated gradient direction $G_{locgrad}$ as $G_d$. Secondly, their angle is greater than 90°, which indicates that the local knowledge conflicts with the general knowledge. In this case, we project the $G_d$ to the orthogonal direction of $G_g$ to optimize the classification model, which avoids increasing the KL loss. Our LocGrad strategy is mathematically formulated as:

$$G_{locgrad} = \begin{cases} G_d, & if \ G_d \cdot G_g \geq 0, \\ G_d - \lambda \cdot \frac{G_d \cdot G_g}{\|G_g\|_2} G_g, & otherwise, \end{cases} \quad (9)$$

where $\lambda$ is a scaling factor for adjusting the projection. Instead of updating the local parameter vectors using $G_d$, we optimize the local parameter vectors using $G_{locgrad}$, which prevents the gradient direction from overfitting to poisoned samples when the participant is malicious.

## 5. Experiment

### 5.1. Experiment Setup

**Dataset**. Following (Li et al., 2021; Mu et al., 2021), we experiment on four federated scenarios.
- MNIST (LeCun et al., 1998) is a handwritten digit dataset of 10 digits class (1$\widetilde{9}$) with $70,000$ images.
- CIFAR-10 (Krizhevsky & Hinton, 2009) has 10 semantics with $50k$, $10k$ images for training, validation.
- CIFAR-100 is a collection of 60,000 32×32 color im-

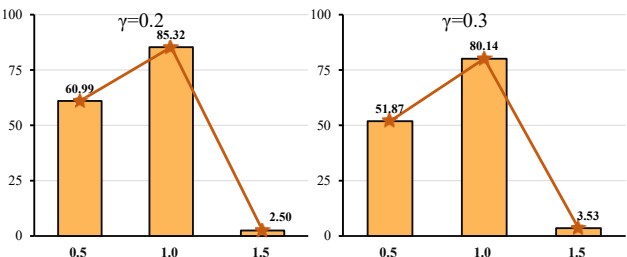

Figure 3. **Comparison of different lambda** $\lambda$ for backdoor failure rate $\mathcal{R}$ on CIFAR-10 with malicious ratio $\gamma = \{0.2, 0.3\}$. We set $\lambda = 1.0$ for the next experiments. See Sec. 5.2.

Table 1. **Ablation on key components** for SPMC in MNIST and CIFAR-10 with $\beta = 0.5$ and $\gamma = \{0.2, 0.3\}$. The $\lambda$ (Eq. 9) for GradReg experiment setting is 1.0. See Sec. 5.2.

| LGAlign Sec. 4.3 | MCAgg Sec. 4.2 | MNIST | | | | | |
|---|---|---|---|---|---|---|---|
| | | $\gamma = 0.2$ | | | $\gamma = 0.3$ | | |
| | | $\mathcal{A}$ | $\mathcal{R}$ | $\mathcal{V}$ | $\mathcal{A}$ | $\mathcal{R}$ | $\mathcal{V}$ |
| | | 99.25 | 2.20 | 50.73 | 99.17 | 1.27 | 50.22 |
| ✓ | | 98.85 | 27.92 | 63.38 | 98.94 | 18.78 | 58.86 |
| | ✓ | 99.27 | 2.11 | 50.69 | 99.19 | 0.52 | 49.85 |
| ✓ | ✓ | 98.79 | 42.73 | **70.76** | 98.72 | 55.95 | **77.34** |

| LGAlign Sec. 4.3 | MCAgg Sec. 4.2 | CIFAR-10 | | | | | |
|---|---|---|---|---|---|---|---|
| | | $\gamma = 0.2$ | | | $\gamma = 0.3$ | | |
| | | $\mathcal{A}$ | $\mathcal{R}$ | $\mathcal{V}$ | $\mathcal{A}$ | $\mathcal{R}$ | $\mathcal{V}$ |
| | | 65.03 | 50.62 | 57.83 | 64.82 | 36.12 | 50.47 |
| ✓ | | 66.66 | 81.90 | 74.28 | 65.13 | 77.57 | 71.35 |
| | ✓ | 64.96 | 48.18 | 56.57 | 63.81 | 37.17 | 50.49 |
| ✓ | ✓ | 66.78 | 85.32 | 76.05 | 65.83 | 80.14 | **72.98** |

ages in 100 classes, with 600 images per class, commonly used for image classification tasks in machine learning and computer vision.

- FashionMNIST (Xiao et al., 2017) is a dataset of 70,000 grayscale images of 10 fashion categories.

As for the data heterogeneity simulation, we utilize the Dirichlet distribution $\beta$ to simulate the label skew, as previous methods (Li et al., 2020a; 2021; Zhang et al., 2022a; Huang et al., 2023c), where $\beta > 0$ is the concentration parameter to adjust the class-wise skew level. We set $\beta = 0.5$ to simulate the heterogeneous federated network for following experiments.

**Counterparts.** We compare three types of backdoor defense solutions. The details are as follows:

**i)** Distance Difference Defense:
- FoolsGold [arXiv'18] (Fung et al., 2018): Analysing similarity of updates to mitigate malicious clients.
- DnC [NDSS'21] (Shejwalkar & Houmansadr, 2021): Downgrading peacekeeping and outlier removal against model poisoning attacks in federated learning.
- Sageflow [NeurIPS'21] (Park et al., 2021): Combine entropy filtering and loss dynamically adjusting weights.
- Finetuning [ToolsforDL'19] (Quinn et al., 2019): Directly optimizes the aggregated global model on vaidation.

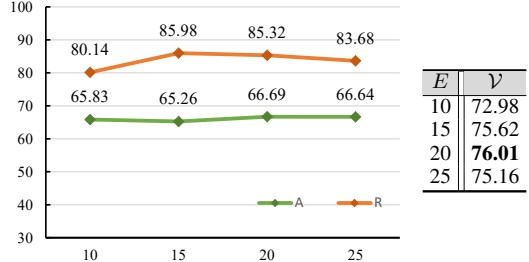

Figure 4. **Ablation on local epoch** $E$ on CIFAR-10 ($\beta = 0.5$) with $\gamma = 0.3$ for the federated benign performance $\mathcal{A}$ (Green line), backdoor failure rate $\mathcal{R}$ (Orange line) and the trade-off $\mathcal{V}$ (Right Table). More details see Sec. 5.2.

| $E$ | $\mathcal{V}$ |
|---|---|
| 10 | 72.98 |
| 15 | 75.62 |
| 20 | **76.01** |
| 25 | 75.16 |

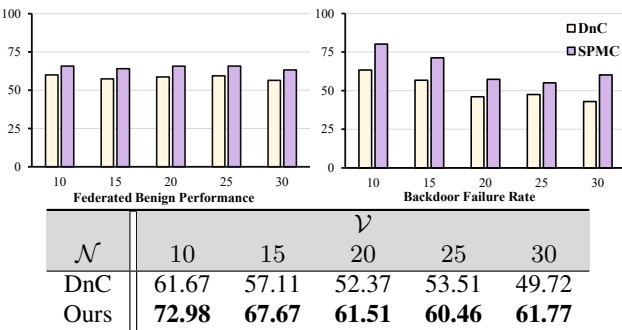

| $\mathcal{N}$ | $\mathcal{V}$ | | | | |
|---|---|---|---|---|---|
| | 10 | 15 | 20 | 25 | 30 |
| DnC | 61.67 | 57.11 | 52.37 | 53.51 | 49.72 |
| Ours | **72.98** | **67.67** | **61.51** | **60.46** | **61.77** |

Figure 5. **Ablation on Client num** $\mathcal{N}$ on CIFAR-10 ($\beta = 0.5, \gamma = 0.3$) with the popular counterpart. See Sec. 5.2.

**ii)** Statistics Distribution Defense:
- Bulyan [ICML'18] (Guerraoui et al., 2018): Agree on coordinates by principal component vectors, chosen via Byzantine–resilient aggregations algorithms.
- RFA [TSP'22] (Pillutla et al., 2022): Leverage the geometric median and the smoothed Weiszfeld algorithm.

**iii)** Objective Optimized Defense:
- RSA [AAAI'19] (Li et al., 2019): Norm regularization and random subgradient aggregation against Byzantine.
- RLR [AAAI'21] (Ozdayi et al., 2021): Adjust the server learning rate with dimension and communication.
- CRFL [ICML'21] (Xie et al., 2021): Exploit clipping and smoothing operations to train federated learning model.

**Backdoor Attacks**. We demonstrate the effectiveness of the proposed method under the popular paradigm (Gu et al., 2019; Han et al., 2023). The size of the trigger pattern $\tau$ is $2 \times 6$ and its location is in the top-left corner of the images. We convert the attacked label to the second class (i.e., 2 in Digits). We set the malicious client ratio $\gamma$ as $\{0.2, 0.3\}$. The local data poisoned portion is default set as 0.3.

**Evaluation Metric**. Given that the backdoored model $f_w$ is anticipated to misclassify poisoned dataset $\widetilde{D}$ as $\widetilde{y}$, the Backdoor Failure Rate $\mathcal{R} = 1 - \frac{1}{\|\widetilde{D}\|} \sum_{x \in \widetilde{D}} \{f_w(x_i) = \widetilde{y}\}$ based on the Attack Success Rate (ASR) (Han et al., 2023; Krizhevsky & Hinton, 2009; Ye et al., 2025a). Additionally, the trained model $f_w$ should produce normal outputs

*Table 2.* **Comparison with the state-of-the-art backdoor robust solutions** in the FashionMNIST, CIFAR-10, and CIFAR-100 dataset with $\gamma \in \{0.2, 0.3\}$. Up arrows ↑ indicate advancements in the given metric compared to FedAvg, while down arrows ↓ denote regressions. The **bolded number** is the best result in the irregular case, and – means failure optimization. Refer to Sec. 5.3 for detailed explanations.

| Methods | FashionMNIST | | | CIFAR-10 | | | CIFAR-100 | | |
|---|---|---|---|---|---|---|---|---|---|
| $\gamma = 0.2$ | $\mathcal{A}$ | $\mathcal{R}$ | $\mathcal{V}$ | $\mathcal{A}$ | $\mathcal{R}$ | $\mathcal{V}$ | $\mathcal{A}$ | $\mathcal{R}$ | $\mathcal{V}$ |
| FedAvg | 87.89 | 4.73 | 46.31 | 65.03 | 50.62 | 57.83 | 57.82 | 15.75 | 36.78 |
| *Predefined Scale Requirement* | | | | | | | | | |
| DnC | 87.25 | 88.70 | 87.97 | 59.79 | 80.93 | 70.36 | 57.52 | 15.91 | 36.72 |
| Sageflow | 88.15 | 9.48 | 48.81 | 64.55 | 51.88 | 58.22 | 57.49 | 17.87 | 37.68 |
| Bulyan | 38.12 | 99.94 | 69.03 | 10.61 | 100.0 | 55.30 | 51.64 | 13.23 | 32.43 |
| RFA | 85.66 | 0.18 | 42.92 | 64.33 | 72.47 | 68.40 | 57.15 | 8.09 | 32.62 |
| RLR | 87.69 | 7.48 | 47.58 | 64.32 | 45.59 | 54.96 | 45.94 | 25.85 | 35.89 |
| CRFL | 84.19 | 1.04 | 42.62 | 49.45 | 64.22 | 56.8 | 26.50 | 60.41 | 43.46 |
| *No Predefined Scale Requirement* | | | | | | | | | |
| FoolsGold | 82.92 | 0.27 | 41.60 | 54.28 | 94.01 | 74.15 | 24.92 | 68.63 | 46.77 |
| RSA | 10.00 | 99.99 | 54.99 | 10.00 | 100.00 | 55.00 | – | – | – |
| Finetuning | 87.15 | 16.71 | 51.93 | 59.70 | 59.17 | 59.44 | 41.43 | 37.93 | 39.68 |
| Ours | 82.19↓5.69 | 70.07↑65.3 | **76.45**↑30.1 | 66.78↑1.75 | 85.32↑34.7 | **76.05**↑18.2 | 64.52↑6.70 | 49.87↑34.12 | **57.20**↑20.42 |

| Methods | FashionMNIST | | | CIFAR-10 | | | CIFAR-100 | | |
|---|---|---|---|---|---|---|---|---|---|
| $\gamma = 0.3$ | $\mathcal{A}$ | $\mathcal{R}$ | $\mathcal{V}$ | $\mathcal{A}$ | $\mathcal{R}$ | $\mathcal{V}$ | $\mathcal{A}$ | $\mathcal{R}$ | $\mathcal{V}$ |
| FedAvg | 88.13 | 0.95 | 44.54 | 64.82 | 36.12 | 50.47 | 56.06 | 11.86 | 33.96 |
| *Predefined Scale Requirement* | | | | | | | | | |
| DnC | 87.09 | 34.49 | 60.79 | 59.99 | 63.35 | 61.67 | 56.14 | 11.01 | 33.58 |
| Sageflow | 87.84 | 0.47 | 44.16 | 64.87 | 36.88 | 50.88 | 56.07 | 11.03 | 33.69 |
| Bulyan | 45.05 | 86.67 | 65.86 | 10.00 | 80.00 | 45.00 | 51.77 | 3.92 | 27.84 |
| RFA | 85.61 | 0.06 | 42.83 | 63.64 | 40.0 | 51.82 | 54.74 | 4.67 | 29.71 |
| RLR | 87.52 | 0.83 | 44.17 | 63.63 | 36.84 | 50.24 | 52.68 | 11.68 | 32.18 |
| CRFL | 78.62 | 0.10 | 39.36 | 45.10 | 49.83 | 47.47 | 25.99 | 57.75 | 41.87 |
| *No Predefined Scale Requirement* | | | | | | | | | |
| FoolsGold | 79.98 | 0.04 | 40.01 | 56.94 | 37.17 | 47.06 | 22.90 | 55.40 | 39.15 |
| RSA | 10.20 | 86.19 | 48.20 | 10.00 | 100.00 | 55.00 | – | – | – |
| Finetuning | 87.16 | 2.48 | 44.82 | 57.36 | 54.11 | 55.74 | 38.60 | 41.07 | 39.83 |
| Ours | 85.14↓2.99 | 60.52↑59.6 | **72.83**↑28.3 | 65.83↑1.01 | 80.14↑44.0 | **72.98**↑22.5 | 66.12↓10.06 | 19.02↑7.16 | **42.57**↑8.61 |

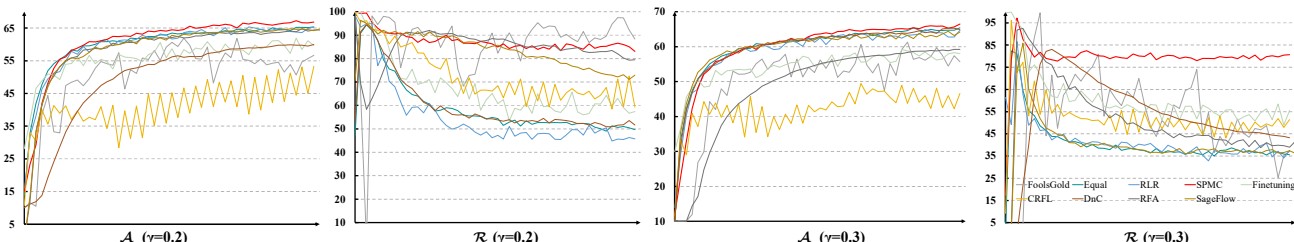

*Figure 6.* **Comparison of federated benign performance** $\mathcal{A}$ **and backdoor failure rate** $\mathcal{R}$ on CIFAR-10 with $\gamma = \{0.2, 0.3\}$. SPMC appears to have stable convergence speed and satisfying performance.

on benign samples (e.g., images without triggers). The model's accuracy on these samples can be measured using the metric called Main Task Accuracy (MTA) (Xie et al., 2020) on benign samples. This is calculated as $\mathcal{A} = \frac{1}{\|D\|} \sum_{x \in D} \{f_w(x_i) = y\}$, where $D$ denotes the validation dataset without malicious trigger pattern. Furthermore, we define the $\mathcal{V}$ to measure the heterogeneity and robustness trade-off as: $\mathcal{V} = \frac{\mathcal{A}+\mathcal{R}}{2}$. All three of these metrics are proportional to model performance. We utilize the mean performance value of the last five communication epochs as the final evaluation results.

**Network Structure**. Following (Huang et al., 2023c; Li et al., 2021), we utilize CNN as the backbone. Specifically, we use a simple CNN structure on FashionMNIST, MNIST, and CIFAR-10. Moreover, we use resnet-18 as the backbone on CIFAR-100 dataset.

**Training Set**. For model efficiency and algorithmic convergence consideration, we conduct communication epochs for $E = 50$ for three datasets. We set local updating round $T = 10$, where all federated learning approaches have little or no accuracy gain with more communications. We use the SGD optimizer with the learning rate $\eta = 0.01$ for all approaches. The corresponding weight decay is $1e - 5$ and momentum is $0.9$. The training batch size is $64$.

### 5.2. Diagnostic Experiments

For a thorough analysis, we perform a set of ablative studies on MNIST and CIFAR-10 with label skew $\beta = 0.5$ and malicious ratio $\gamma$ selected by $\{0.2, 0.3\}$.

**Control Hyper-Parameter** $\lambda$ in Eq 9. Fig. 3 quantifies the effect of the hyperparameter $\lambda$, which measures the strength

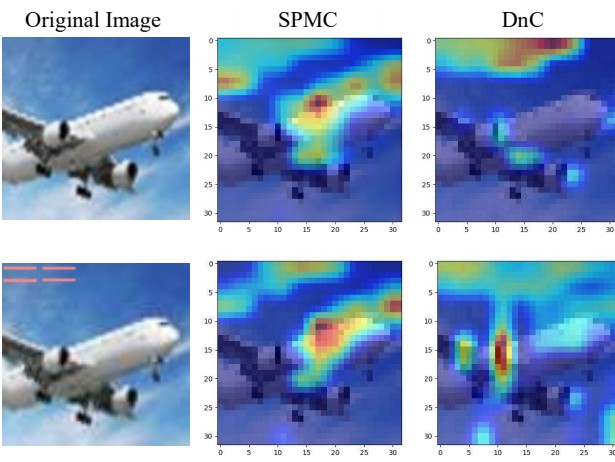

*Figure 7.* **Comparison of heat maps** for SPMC and DnC with and without the trigger. Final models are trained with $\gamma = 0.3$.

of the gradient alignment in different scenarios. Specifically, the proportion of malicious attackers does not affect the effect of the hyperparameter $\lambda$ on the model. As can be seen from the figure, the effect is not significant when $\lambda$ exceeds 1.0 or is less than 1.0. This is because when the alignment criterion is less than $\lambda = 1.0$, some benign participants are forced to learn a part of the malicious knowledge. If the alignment criterion is greater than $\lambda = 1.0$, the malingerers are prone to keep the original update direction. As a result, under strict regularization stiffness, the alignment gradient is effective for method improvement. In the next experiments, we choose $\lambda = 1.0$ for different scenarios.

**Target Objective**. We quantitatively analyze the proposed SPMC. In Table 1, although Local Gradient Alignment (LGAlign) has a good performance, it still performs poorly with a high percentage of malicious clients with complex images as input. Even in the case of a high percentage of malicious clients, combining Local Gradient Alignment (LGAlign) and Margin Contribution Aggregation (MCAgg) acquires satisfying federated benign task and backdoor removal performance, which is consistent with the fact that we utilize the margin coalition model for Local Gradient Alignment and Server Aggregation motivation.

**Local Updating Rounds**. Furthermore, we analyze the effect of local updating rounds in Fig. 4. SPMC maintains stable performance under different local rounds, indicating that SPMC achieves fast convergence in limited epochs and possesses the ability to resist client drift under various local rounds (Zhang et al., 2023; Moreno-Torres et al., 2012). We choose $T = 10$ as the updating round.

**Client Scale** $\mathcal{N}$. We evaluate the performance with various participating client scales $\mathcal{N}$ in Fig. 5. Our SPMC achieves competitive heterogeneity and robustness trade-off performances, demonstrating that our method is capable of dealing with the high client scale in the federated system.

## 5.3. Comparison to State-of-the-Arts

Tab. 2 plots the final metric by the end of the federated learning process with popular backdoor defense methods. Despite the handwritten dataset with simple features, SPMC demonstrates robustness when compared to other methods vulnerability on handwritten datasets. Moreover, although DnC demonstrates superior performance in scenarios with a low attacker ratio ($\gamma = 0.2$), the limitations of the pre-defined rules become evident as the number of attackers varies. It shows that our method achieves a more satisfying performance than our different counterparts on different evaluation metrics, which confirms that SPMC effectively enhances the backdoor robustness in heterogeneous federated learning. Take the result of CIFAR-100 with $\gamma = 0.2$ as an example, our method outperforms the best counterpart with a gap of $10.43\%$ on the $\mathcal{V}$ metric. Furthermore, existing backdoor defensive methods appear fragile Backdoor Failure Rate $\mathcal{R}$ under either the large malicious client ratio $\gamma = 0.3$. It reveals that existing solutions fail to conduct client-wise discrimination selection under large-scale evils. The expanded experiment on the MNIST dataset is shown in Appendix C.2.

We further plot both the Federated Benign Performance $\mathcal{A}$ and Backdoor Failure Rate $\mathcal{R}$ during the communication process on CIFAR-10 setting in Fig. 6. We observe that SDFC presents faster and stabler convergence speed than others with different malicious ratios. We utilize Grad-CAM (Selvaraju et al., 2017; He et al., 2023) to visualize the network attention with or without the trigger (Fan et al., 2025). Compared to DnC, SPMC prefers to extract key features despite attacks (Fig. 7). We visualize the learned features using t-SNE in Appendix C.3. Results shows SPMC effectively purifies poisoned samples, mitigating their influence and substantially lowering the attack success rate.

## 6. Conclusion

We present self-purifying backdoor defense via margin contribution (SPMC), the first work to implement backdoor robustness in heterogeneous federated learning based on margin contributions. We argue that existing backdoor defenses either focus only on individual behaviors, rely on predefined scales, or utilize proxy data to design well-designed client selection mechanisms. However, malicious clients can adapt their adversarial strategies to bypass these defense protocols. Therefore, we utilize margin contributions to quantify the discrepancy between the local parameter distribution and its coalition model parameter distribution, enabling the system to self-purify malicious impacts. The effectiveness and robustness of our method are validated against common backdoor attacks involving varying numbers of adversaries. We hope this work offers a novel perspective and paves the way for future research in this domain.

## Acknowledgement

This work is supported by the National Natural Science Foundation of China under Grant (62361166629, 62176188, 623B2080), the Hubei Provincial Natural Science Foundation of China (2025AFB219). The supercomputing system at the Supercomputing Center of Wuhan University supported the numerical calculations in this paper.

## Impact Statement

This paper presents work whose goal is to advance the field of Machine Learning. There are many potential societal consequences of our work, none of which we feel must be specifically highlighted here.

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

# APPENDIX

## A. Notation Table

We provide the notation table in Tab. 3.

*Table 3.* **Notations** table.

| | Description | | Description |
|---|---|---|---|
| $k$ | Client index | $N$ | All clients gather |
| $D_k$ | $k^{th}$ client private data | $\widetilde{D}_k$ | $k^{th}$ client poisoned data |
| $\widetilde{x}$ | Data with trigger | $\widetilde{y}$ | Poisoned target |
| $w_g$ | Shared global network | $I_k$ | Model parameter distribution |
| $\alpha_k$ | $k^{th}$ client's aggregation weight | $t$ | Communication round index |
| $\eta$ | Local learning rate | $S_k$ | The coalition of $k^{th}$ client |
| $\phi_k$ | The shapley value | $\Gamma(k)$ | The value of $k^{th}$ client |
| $\sigma$ | The function of softmax | $G_{locagrad}$ | Local update direction |
| $G_d$ | The direction of $\mathcal{L}_{CE}$ | $G_g$ | The direction of $\mathcal{L}_{kl}$ |

## B. Algorithm

We provide the algorithm description in Algorithm 1.

---
**Algorithm 1** SPMC
---
**Data:** The local dataset $D_k$
**Input:** Communication rounds $T$, participant set $N$, $k^{th}$ client private model $I_k$, learnable aggregation weight $\alpha_k$ with updating epoch $E$ and learning rate $\eta$
**Output:** The final global model $w_g$

**for** $t = 1, 2, ..., T$ **do**
  *Participant Side*
  **for** $k = 1, 2, ..., K$ in parallel **do**
    $I_k^t \leftarrow \text{LocalUpdating}(w^t, \alpha_k)$
  **end**
  *Server Side*
  // Reweight by margin contribution
  $\{\alpha_k\}_{k=1}^N \leftarrow$ Eq. (6)
  $w_g^{t+1} \leftarrow \sum_{k=1}^N \alpha_k I_k^t$
**end**

LocalUpdating($w_g^{t-1}, \alpha_k$):
// Calculate the coalition model
$I_{S_k} \leftarrow$ Eq. (5)
**for** $e = 1, 2, ..., E$ **do**
  **for** $B_g = \{x_i\} \subset D_g$ **do**
    // Calculate local updated direction by projection
    $G_d = \nabla \mathcal{L}_{CE}$
    $G_g = \nabla \mathcal{L}_{kl}$
    $G_{locagrad} \leftarrow$ Eq. (9)
    // Local Updating
    $I_k^{t+1} \leftarrow w_k^t - \eta G_{locagrad}$
  **end**
**end**
---

## C. More experiments

### C.1. The choice of metric on Margin Contribution

We used cosine similarity, Euclidean distance, and Wasserstein distance as evaluation metrics to measure $\Gamma(N\{n\}) - \Gamma(\{n\})$. As shown in Table 5, cosine similarity proves to be the most suitable metric. In SPMC, cosine similarity

is chosen primarily because it effectively measures the directional difference between two vectors, rather than their magnitude (as in Euclidean distance) or the divergence between probability distributions (as in Wasserstein distance). In the model parameter space, this allows for the evaluation of directional consistency among client updates, which is particularly important for identifying malicious attackers. Malicious clients often train their local models on poisoned datasets, resulting in update directions that deviate significantly from those of benign clients.

*Table 4.* Comparison with evaluation metrics on the CIFAR-10 dataset with malicious proportion $\gamma$=0.3. We use the trade-off V to evaluate the performance of different evaluation metrics.

| | Cosine similarity | Euclidean distance | Wasserstein distance |
|---|---|---|---|
| SPMC | **72.98** | 45.30 | 69.09 |

### C.2. Expanded experiment on MNIST dataset

*Table 5.* Comparison with the state-of-the-art backdoor robust solutions in the MNIST dataset. Up arrows ↑ indicate advancements in the given metric compared to FedAvg, while down arrows ↓ denote regressions. The **bolded number** is the best result.

| Method | $\gamma = 0.2$ | | | $\gamma = 0.3$ | | |
|---|---|---|---|---|---|---|
| | $\mathcal{A}$ | $\mathcal{R}$ | $\mathcal{V}$ | $\mathcal{A}$ | $\mathcal{R}$ | $\mathcal{V}$ |
| FedAvg | 99.25 | 2.20 | 50.73 | 99.17 | 1.27 | 50.22 |
| *Predefined Scale Requirement* | | | | | | |
| DnC | 99.01 | 77.77 | 88.39 | 99.07 | 1.62 | 50.34 |
| Sageflow | 99.21 | 1.69 | 50.45 | 99.29 | 2.21 | 50.75 |
| Bulyan | 10.54 | 100.0 | 55.27 | 10.31 | 60.0 | 35.15 |
| RFA | 99.09 | 0.26 | 49.68 | 99.27 | 0.15 | 49.71 |
| RLR | 99.07 | 1.71 | 50.39 | 99.11 | 0.91 | 50.01 |
| CRFL | 97.87 | 3.01 | 50.38 | 97.60 | 0.36 | 48.98 |
| *No Predefined Scale Requirement* | | | | | | |
| FoolsGold | 96.13 | 0.37 | 48.25 | 81.56 | 9.48 | 45.52 |
| RSA | 30.25 | 88.18 | 59.22 | 35.29 | 79.70 | 57.49 |
| Finetuning | 98.89 | 3.88 | 51.38 | 98.88 | 2.34 | 50.61 |
| Ours | 98.79$_{\downarrow0.46}$ | 42.73$_{\uparrow40.5}$ | **70.76**$_{\uparrow20.0}$ | 98.72$_{\downarrow0.45}$ | 55.95$_{\uparrow54.6}$ | **77.33**$_{\uparrow22.1}$ |

### C.3. Visualization of Feature Distributions on CIFAR-10

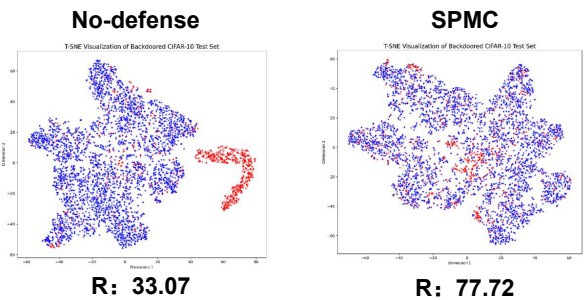

**No-defense**      **SPMC**

R: 33.07      R: 77.72

*Figure 8.* T-SNE visualization of the feature representations on the CIFAR-10 dataset. Blue point corresponds to benign samples and red point represents poisoned samples. In the absence of defense, the poisoned samples are clearly distinguishable and form clusters separate from the benign data, indicating that the backdoor attack successfully manipulates the feature space to achieve high attack success rates. However, after SPMC, these poisoned features become less distinguishable, blending into the benign distribution.

