# OpenReview forum: "SPMC: Self-Purifying Federated Backdoor Defense via Margin Contribution"
_ICML.cc/2025/Conference — ICML 2025 poster_

### Official Review · Reviewer_7qEv · 2025-03-06

**Overall Recommendation:** 3

**Summary:**

The authors proposed a new federated backdoor defense method named SPMC. SPMC employs Shapley value to assess the contribution of each client and then reweights their model updates (server side). Additionally, SPMC utilizes knowledge distillation to calibrate the direction of gradients (client side). The authors conducted several experiments to validate SPMC.

**Claims And Evidence:**

Some claims lack formal substantiation. For instance, it is unclear why the federation resulting from a new client joining the existing client pool would produce a malicious contribution (right column, Lines 196-198).

**Essential References Not Discussed:**

The authors have missed some federated backdoor attack methods [1].

[1] Bad-PFL: Exploring Backdoor Attacks against Personalized Federated Learning, ICLR

**Experimental Designs Or Analyses:**

The experiments are flawed as the authors have overlooked many recent backdoor attack methods.

**Methods And Evaluation Criteria:**

The authors introduce a new method. This paper does not involve any new datasets.

**Other Comments Or Suggestions:**

* It would be beneficial to present experimental results demonstrating the proposed method's effectiveness against more advanced attack methods.

* Since FedAvg is hardly used in Non-IID scenarios, I recommend the authors consider incorporating some personalized federated learning methods.

**Other Strengths And Weaknesses:**

The paper is well-written and easy to follow. The paper has a good motivation and intuition, but I believe the experiments are insufficient. The authors only validated the effectiveness of their method against DBA.

**Questions For Authors:**

See above.

**Relation To Broader Scientific Literature:**

The proposed method is a new federated backdoor defense method, which could potentially enhance the robustness of federated learning.

**Theoretical Claims:**

The paper does not contain theoretical claims.

---

> ### Author Rebuttal · Authors · 2025-03-29
>
> Dear Reviewer 7qEv：
>
> Thank you for your thoughtful review and for raising key concerns regarding our work. We hope the following responses will address your concerns and update the score.
>
> **Question**
>
> **Q1: Lack of substantiation and theoretical claims.** (Claims And Evidence&Theoretical Claims)
>
> A1: We sincerely apologize for the confusion. Inspired by the Shapley value, we define a client's contribution to its marginal coalition as the marginal contribution. Specifically, for client $k$, its contribution to the marginal coalition is denoted as $\phi_k(k;\ \Gamma;\ D;\ N)=\mathbb{E}_{\Gamma\sim D^N}[\Gamma(N\setminus{k})-\Gamma({k})]$，where $D^N$ represents the collection of all clients' datasets, and $\Gamma(N\setminus{k})-\Gamma({k})$ captures the model parameter difference between the marginal coalition and client k. Next, we introduce the theorem to demonstrate the role of marginal contribution in identifying malicious clients.
> Theorem: Let parameter aggregation function $\Gamma$ be B-Lipschitz with respect to $Z$. Suppose $D_k$ and $D_p$ are two data distribution over $Z$. Then, for client $k,p\in N$,
> $$
> \phi_k(k;\ \Gamma;\ D_k;\ N)-\phi_p(p;\ \Gamma;\ D_p;\ N)\le2NB·W(D_k,D_p),
> $$
> where $W$ denotes the distributional difference between two data distributions. This theorem measures the marginal contribution changes under two different data distributions. Due to the significantly different distribution of malicious clients, the marginal contributions of malicious clients become easier to distinguish. We will add this theorem's derivation in final version. Thank you for the valuable comment.
>
> **Q2: Extended experiments on a new dataset.** (Methods And Evaluation Criteria)
>
> A2: We agree with you and have incorporated this suggestion throughout our paper. We extended our experiments on the CIFAR-100 dataset using ResNet-18. As shown in Table 1, even in more complex task scenarios, both DnC and SPMC maintain their defensive effectiveness. However, DnC struggles to match SPMC’s backdoor defense due to the limited information captured by its sub-dimensions. Moreover, the increased model complexity and task difficulty hinder convergence of the local gradient learning rate, significantly impacting RLR's main task performance.
>
> *Table 1: Comparison with the SOTA backdoor robust solutions on CIFAR-100 dataset using ResNet-18. The malicious proportion is  γ=0.3 and the local data poisoned portion is set as 0.3.*
>
> | Method | A↑| R↑| V↑|
> |-----|-----|----- |-----|
> |Fedavg|55.29|6.91|31.10|
> |DnC|54.44|11.31|32.88|
> |RLR| 50.94|7.86|29.40|
> |Ours| 55.14|35.26|**45.20**|
>
> **Q3: Lack of comparison to recent backdoor attack method under PFL.** (Essential References Not Discussed&Other Comments Or Suggestions&Other Strengths And Weaknesses)
>
> A3: Thank you for your suggestion. We would like to clarify that the attack method used in our approach is BadNet, not DBA. BadNet aims to poison the model by injecting whole static triggers into the local dataset. In contrast, DBA constructs a global trigger by combining local triggers across multiple clients.
>
> To verify that SPMC can maintain the robustness of federated systems in the stronger attack, we followed your suggestion and introduced an attack method, Bad-PFL [1]. This method leverages natural data features as triggers, achieving both effectiveness and stealth in PFL method (i.e., FedProx [2]). As shown in Table 2, BadPFL achieves better attack performance compared to BadNet in PFL.
>
> *Table 2: Comparison with BadNet and Bad-PFL under PFL. The experiment is based on CIFAR-10 with malicious ratio of 0.3 and SimpleCNN.*
>
> ||Fedavg||Fedprox||
> |:-----:|:-----:|:-----:|:-----:|:-----:|
> |Method|A↑|R↓|A↑|R↓|
> |BadNet|64.82|36.12|61.37|35.03|
> |Bad-PFL|**70.08**|**8.65**|**68.14**|**8.17**|
>
> Table 3 presents the resistance of two defense methods and one no-defense method (Equal) against the BadPFL attack in the context of personalized federated learning.
>
> *Table 3: Comparison with the SOTA backdoor defense with **Bad-PFL** attack under the PFL. The experiment is based on CIFAR-10 dataset with malicious ratio of 0.3 and SimpleCNN.*
>
> | | | Fedavg | | | Fedprox | |
> |:----:|:---:|:----:| :-----: | :---: | :-----: | :-----: |
> |Method|A↑|R↑|V↑| A↑|R↑|V↑|
> |Equal|70.08|8.65|39.37|68.14|8.17|38.16|
> |DnC|61.95|62.86|62.51|60.13|53.77|56.95|
> |Ours|67.72|71.87|**69.80**|58.65|61.58|**60.12**|
>
> As shown in Table 3, although BadPFL achieves robust attack performance in personalized federated learning settings, it can still be effectively detected by defense methods such as DnC and SPMC. Moreover, SPMC demonstrates better defense effectiveness compared to DnC. Overall, even in the face of advanced attack strategies and PFL method, SPMC is able to maintain the security of the federated system to a considerable extent.
>
> [1] Bad-pfl: Exploring backdoor attacks against personalized federated learning.In Proc. of ICLR, 2025.
>
> [2] Federated optimization in heterogeneous networks. In MLSys, 2020a.

---

### Official Review · Reviewer_VbBm · 2025-03-09

**Overall Recommendation:** 3

**Summary:**

This paper introduces a technique named SPMC (Self-Purifying Federated Backdoor Defense via Margin Contribution), which aims to detect and mitigate backdoor attacks in federated learning systems by leveraging the concept of Shapley values. SPMC not only focuses on the behavior of individual clients but also emphasizes the interactions among clients and their impact on the overall coalition model.

**Claims And Evidence:**

Yes

**Essential References Not Discussed:**

No

**Experimental Designs Or Analyses:**

The experimental metrics are acceptable, but lacks support from other experiments.

**Methods And Evaluation Criteria:**

Yes

**Other Comments Or Suggestions:**

The elements in the framework diagram of the article are somewhat redundant.

**Other Strengths And Weaknesses:**

The authors categorize existing defense methods into three types and summarize that these methods are primarily based on two assumptions. They provide a very clear motivation and introduce SPMC, a method that leverages margin contributions to defend against backdoor attacks. The paper clearly explains how SPMC is deployed on both the client and server sides, and experiments demonstrate the effectiveness of SPMC. The major strengths of this work are as follows:
- The author's summary of the federated defense methods is very interesting. The motivation of the article is clear.
- Experimental results (Figure 7) have shown that SPMC can extract key features from the image even though the attackers add triggers to images.

Weakness

- The experimental results on traditional datasets like CIFAR-10 and MNIST demonstrate promising performance of SPMC. However, CIFAR-10 and MNIST are commonly used simple datasets. Have the authors attempted testing on complex datasets to further validate the effectiveness of SPMC?
- The article seems to only discuss the scenarios with attack ratios of 0.2 and 0.3. I am curious to know what would happen when the attack ratio is set to 0.5.
- I am still a bit confused about the concept of margin contribution. Can you briefly explain the concept of "margin contribution" as used in this paper?
- The color contrast in the article is not strong enough, making it difficult for me to focus on the key points in the figures.

**Questions For Authors:**

No

**Relation To Broader Scientific Literature:**

SPMC not only defends against backdoor attacks at the server side but also prevents malicious updates locally, thereby extending the robustness of federated learning systems beyond traditional server-centric defenses.

**Theoretical Claims:**

SPMC provides a comprehensive summary of federated defense methods and clearly describes the motivation and details of the proposed method.

---

> ### Author Rebuttal · Authors · 2025-03-30
>
> Dear Reviewer VbBm：
>
> Thank you very much for your valuable suggestions—they have provided us with clear direction for further exploring the details of our method. We hope the following responses will address your concerns.
>
> **Question**
>
> **Q1: Lack of support from other experiments.** (Experimental Designs Or Analyses)
>
> A1: We will expand to more complex experimental settings and incorporate a wider range of attack types in the final version. Thank you for your suggestion.
>
> **Weakness**
>
> **W1:** **Evaluation on complex datasets.**
>
> A2: We agree with you and have incorporated this suggestion throughout our paper. We have extended our experiments on the CIFAR-100 dataset using ResNet-18 as the backbone. As shown in Table 1, even in more complex models and task scenarios, both DnC and SPMC are able to maintain their defensive effectiveness. However, DnC struggles to achieve a stronger backdoor performance compared to SPMC because of the poor sub-dimensions' information. In addition, due to the increased model complexity and the difficulty of classification tasks, the gradient learning rate on local clients fails to converge, which significantly undermines the RLR's performance of the main task.
>
> *Table 1: Comparison with the state-of-the-art backdoor robust solutions on CIFAR-100 dataset using ResNet-18 as the backbone. The malicious proportion is γ=0.3, and the local data poisoned portion is set as 0.3.*
>
> | Method | A↑| R↑| V↑|
> | ------ | ----- | ----- | --------- |
> | Fedavg|55.29|6.91|31.10|
> | DnC|54.44|11.31|32.88|
> | RLR| 50.94|7.86|29.40|
> | Ours| 55.14|35.26|**45.20**|
>
> **W2: Discussion the scenario with attack ratio set to 0.5.**
>
> A3: You have raised an interesting concern. We can compare the performance of SPMC under high attack ratio (i.e, γ=0.5) and low attack ratio (i.e, γ=0.3). As shown in Table 2, when the proportion of malicious clients increases to 0.5, it becomes more challenging to identify and mitigate the backdoor attack, as the probability that malicious clients have marginal contributions similar to those of the marginal coalition significantly increases. However, we believe that high attack ratio would be outside the scope of our paper. In real-world deployments of federated learning systems, the proportion of attackers is typically low [1]. Therefore, we focus on attack ratios of 0.2 and 0.3, which are more consistent with commonly observed threat models in practice. In this work, SPMC demonstrates reliable defense performance across multiple task settings, indicating its effectiveness in maintaining the robustness of real-world federated systems.
>
> *Table 2: Comparison with different malicious proportion γ on CIFAR-10 dataset with ResNet-18.*
>
> |        |             |     0.3     |             |             |     0.5     |             |
> | :----: | :---------: | :---------: | :---------: | :---------: | :---------: | :---------: |
> |        | A↑ | R↑ | V↑| A↑ | R↑ | V↑ |
> | Fedavg |    89.77    |    13.19    |    51.48    |    89.87    |    12.86    |    51.37    |
> |  Ours  |    90.32    |    31.69    |  **61.01**  |    87.83    |    18.68    |  **53.26**  |
>
> **W3: Explanation the concept of "margin contribution".**
>
> A4: Our method is inspired by the Shapley Value, using marginal contribution to measure each client’s impact on the overall federation in federated learning. In the SPMC approach, we estimate a client’s marginal contribution by simplifying the calculation of its influence on the coalition formed by the remaining clients. The server then reallocates aggregation weights based on these contributions, increasing the influence of benign clients while reducing that of malicious ones. On the client side, marginal contribution is used to guide local gradient adjustments, ensuring the update direction aligns with global knowledge and preventing interference from malicious data. This approach effectively identifies and mitigates the impact of attackers, enhancing the security and robustness of the federated learning system. We will add the detailed explanation this in our final version. Thank you for the comment.
>
> **W4: The color contrast in the article is not strong enough.**
>
> A5: We will make the necessary adjustments in the corresponding sections. Thank you for your feedback.
>
> [1] Back to the drawing board: A critical evaluation of poisoning attacks on federated learning. arXiv abs/2108.10241.

---

### Official Review · Reviewer_7GF4 · 2025-03-10

**Overall Recommendation:** 4

**Summary:**

Existing defenses rely on assumptions like individual behavior isolation and passive purification, which malicious clients can bypass. This paper proposes SPMC, inspired by the Shapley Value. It measures inter-client margin contributions to identify malicious attackers and self-purifies the parameter distribution of potential malicious actors. This is achieved through margin contribution aggregation on the server side and local gradient alignment on the client side.

**Claims And Evidence:**

Yes. The authors categorize related work and introduce the crucial reference to induce their claims.

**Essential References Not Discussed:**

N/A

**Experimental Designs Or Analyses:**

The experiments are well-executed and comprehensive, and reasonable explanations are provided for the experimental results.
But the main results are summarized from small datasets and backbones, which could be evaluated in larger and more complex scenarios.

**Methods And Evaluation Criteria:**

The method is well evaluated with clear criteria.

**Other Comments Or Suggestions:**

None

**Other Strengths And Weaknesses:**

Strengths
1 SPMC draws inspiration from the Shapley value by employing inter-client margin contributions to effectively identify malicious attackers. This approach enables a more robust and accurate detection mechanism within federated learning environments.
2 Furthermore, the experiments conducted are comprehensive, offering thorough and reasonable explanations for the observed outcomes, which validate the effectiveness of SPMC. The comparison between various defenses in different directions is interesting, showing that SPMC applies in many scenarios.

Weakness
1 The self-purification part in Figure 2 seems to be unclear, which confuses me about the details of local updates. Could you explain the content of this module to me?
2 What kinds of triggers did the authors use in the experiments?
3 The criteria for selecting Coalitions are not provided in the article; does the choice of Coalitions affect the communication cost?
4 The complex extension to larger datasets and models is less explored and discussed in current version.

**Questions For Authors:**

See above.

**Relation To Broader Scientific Literature:**

This article innovatively applies the Shapley value to defend against backdoor attacks in federated learning, contributing to the broader scientific literature on enhancing distributed system security.

**Theoretical Claims:**

This article utilizes the Shapley value to defend against backdoor attacks. It needs further exploration of the Shapley value theoretical description in future work.

---

> ### Author Rebuttal · Authors · 2025-03-29
>
> Dear Reviewer 7GF4：
>
> We sincerely thank you for the valuable comments and suggestions. We hope our responses below address your concerns and provide a clearer understanding of our approach and results.
>
> **Question**
>
> **Q1: Theoretical** **description for shapley value.** (Theoretical Claims)
>
> A1: Inspired by the concept of the Shapley value, we define a client’s contribution to its marginal coalition as its marginal contribution. For client $k\in\ N$, the marginal contribution is denoted as $\phi_k(k;\ \Gamma;\ D;\ N)=\mathbb{E}_{\Gamma\sim D^N}[\Gamma(N\setminus{k})-\Gamma({k})]$，where $D^N$ represents the collection of all clients' datasets, and $\Gamma(N\setminus{k})-\Gamma({k})$ captures the model parameter difference between the marginal coalition and local client. We introduce the theorem to demonstrate the role of marginal contribution in identifying malicious clients.
>
> Theorem: Let parameter aggregation function $\Gamma$ be B-Lipschitz with respect to Z. Suppose $D_k$ and $D_p$ are two data distribution over $Z$. Then, for client $k,p\in N$,
> $$
> \phi_k(k;\ \Gamma;\ D_k;\ N)-\phi_p(p;\ \Gamma;\ D_p;\ N)\le2NB·W(D_k,D_p),
> $$
> where $W$ denotes the distributional difference between two data distributions. This theorem measures the marginal contribution changes under two different data distributions. Due to the significant different distribution of malicious clients, the marginal contributions of malicious clients become easier to distinguish.
>
> **Q2: Evaluation on large-scale datasets and complex backbone.** (Experimental Designs Or Analyses&Other Strengths And Weaknesses)
>
> A2: Based on your suggestion, we have extended our experiments on the CIFAR-100 dataset using ResNet-18 as the backbone. As shown in Table 1, even in more complex models and task scenarios, both DnC and SPMC are able to maintain their defensive effectiveness. However, DnC struggles to achieve a stronger backdoor performance compared to SPMC because of the poor sub-dimensions' information. In addition, due to the increased model complexity and the difficulty of classification tasks, the gradient learning rate on local clients fails to converge, which significantly undermines the RLR's performance of the main task.
>
> *Table 1: Comparison with the state-of-the-art backdoor robust solutions on CIFAR-100 dataset using ResNet-18 as the backbone. The malicious proportion is  γ=0.3 and the local data poisoned portion is set as 0.3.*
>
> | Method | A↑| R↑| V↑|
> | ------ | ----- | ----- | --------- |
> | Fedavg|55.29|6.91|31.10|
> | DnC|54.44|11.31|32.88|
> | RLR| 50.94|7.86|29.40|
> | Ours| 55.14|35.26|**45.20**|
>
> **Weakness**
>
> **W1: Explanation for the self-purification part of local updates.**
>
> A3: Thank you for the constructive comments. We believe it is important to emphasize the importance of the self-purification. Malicious clients may inject trigger patterns $\tau$ to alter original labels and distort gradient directions, deviating from those of benign clients. To counter this, SPMC introduces a self-purification update mechanism that adjusts local gradients using Equation (9), aligning them with the marginal federation model's knowledge while preserving useful benign local information;  As shown in Table 1 of our paper, the self-purification component of local updates effectively preserves benign information and achieves strong defense performance.
>
> **W2: What kinds of triggers did the authors use in the experiments?**
>
> A4: We use a trigger pattern located at the top-left corner of the image with a size of 2 × 6. The poisoned label is set to class 2. Thank you for your feedback and we will emphasize this point in the final version.
>
> **W3: The criteria for selecting Coalitions and the communication cost for the choice of Coalitions**.
>
> A5: We thank the reviewer for pointing out this issue. We have added the detailed clarification in our revised manuscript. In fact, each client is associated with a corresponding marginal coalition. For example, for client i, its marginal coalition $S_k$ consists of all online clients excluding client $i$. Therefore, we do not need to spend significant computational resources to select the marginal coalition. As shown in the table 3, SPMC does not introduce significant computational cost on either the server or client side. We will update the detailed discussion in the final version.
>
> *Table 3: Computation cost comparison. $n$ refers to the number of online client, $|w|$ represents the scale of network, $|w|_{sub}$ represents the scale of sub-network and $E$ indicates the iteration times of using SVD.*
>
> | Method |Server-side |Client-side|
> | ------ | :------------------------: | :----------------: |
> |Fedavg|$\mathcal{O}(n\times\|w\|\)$|$\mathcal{O}(\|w\|)$|
> |DnC|$\mathcal{O}(n\times\|w\|+E \times\|W\|_{sub}) $ | $\mathcal{O}(\|w\|)$|
> |RLR|$\mathcal{O}(n\times\|w\| )$|$\mathcal{O}(\|w\| )$|
> |Ours|$\mathcal{O}(n\times\|w\|+n^2)$|$\mathcal{O}(\|w\|$)|

---

> > ### Comment · Reviewer_7GF4 · 2025-04-04
> >
> > Thank you for your response. The extra explanations clarify a lot, e.g., extension to large-scale dataset and complex backbone verify the stability of SPMC. I would like to keep my rating unchanged.

---

### Official Review · Reviewer_hJgY · 2025-03-13

**Overall Recommendation:** 4

**Summary:**

This paper presents a federated backdoor defense method named SPMC, which applies Shapley values to quantify the contribution differences among clients and implements margin contribution-based aggregation at the server side and gradient alignment technology at the client side. These measures work together to effectively improve the robustness and flexibility of federated learning systems, even when the number of attackers dynamically changes. Experimental results show that SPMC demonstrates superior defensive effects on multiple public datasets compared to existing methods.

**Claims And Evidence:**

Yes.

**Essential References Not Discussed:**

None

**Experimental Designs Or Analyses:**

The experimental metrics are very comprehensive, with experiments demonstrating the robustness of the method under multiple settings.

**Methods And Evaluation Criteria:**

Yes. The methods are well evaluated.

**Other Comments Or Suggestions:**

See weakness

**Other Strengths And Weaknesses:**

Strengths:
1. The authors effectively highlight the challenges posed by adaptive malicious clients and the need for more flexible and robust solutions. Then, the authors propose SPMC, enabling self-purification without the need for predefined rules.

2. SPMC is novel and introduces an innovative approach to enhancing server-side aggregation and local gradient alignment. It’s easy to follow on other scenarios in the distributed system.

3. The article implements non-proxy distillation, providing a new approach to ensuring the security of proxy datasets in knowledge distillation.

Weakness:

1. Although the experimental results in Table 1 demonstrate the effectiveness of combining the LGAlign and MCAgg modules, the author should present some discussion about why the combination of LGAlign and MCAgg leads to better performance.

2. From the equation (6), the author adopts cosine similarity to measure Γ (N\{n}) − Γ ({n}). The author needs to explain the reasoning behind using cosine similarity instead of other metrics. Are there any better metrics to measure this?

3. In Figure 2, non-proxy distillation seems to be related to the angle. But the article does not provide an intuitive explanation for the values of the hyperparameter λ.

If the author can address all of my concerns, I am willing to increase my score. However, if the author does not address these concerns, I may consider lowering the score.

**Questions For Authors:**

See weakness

**Relation To Broader Scientific Literature:**

This paper innovatively applies Shapley values to proxy-free distillation, addressing the security issues associated with proxy datasets.

**Theoretical Claims:**

This paper provides a clear theoretical explanation of proxy-free distillation and utilizes the framework diagram to explain how to apply Shapley values.

---

> ### Author Rebuttal · Authors · 2025-03-30
>
> Dear Reviewer hJgY：
>
> Thank you for your valuable feedback and for your time reviewing our work. We hope our responses below help clarify the issues and update the score.
>
> **Weaknesses**
>
> **W1: Missing an explanation why the combination of LGAlign and MCAgg leads to better performance.**
>
> A1: For our method, LGAlign represents the local update process. Specifically, it ensures that the update gradients of local clients remain aligned with the general knowledge of the edge federation, thereby preventing attackers from fitting malicious outcomes. As for MCAgg, it represents the server aggregation process. It encourages the positive influence of clients that contribute significantly to the edge federation, while suppressing the malicious impact of those with relatively small marginal contributions. The combination of LGAlign and MCAgg not only prevents local malicious gradients from steering the global update, but also promotes the influence of benign clients through inter-client interactions on the server side. We have rewritten this part to be more in line with your comments. We hope that the edited section clarifies the reason for the combination of LGAlign and MCAgg.
>
> **W2: The reason for choosing cosine similarity to measure Γ (N{n}) − Γ ({n}).**
>
> A2: Thank you for your helpful feedback. We have included a new table to further illustrate the effectiveness of cosine similarity. We used cosine similarity, Euclidean distance, and Wasserstein distance as evaluation metrics to measure Γ (N{n}) − Γ ({n}). As shown in Table 1, cosine similarity proves to be the most suitable metric. In SPMC, cosine similarity is chosen primarily because it effectively measures the directional difference between two vectors, rather than their magnitude (as in Euclidean distance) or the divergence between probability distributions (as in Wasserstein distance). In the model parameter space, this allows for the evaluation of directional consistency among client updates, which is particularly important for identifying malicious attackers. Malicious clients often train their local models on poisoned datasets, resulting in update directions that deviate significantly from those of benign clients. We will supplement this experiment along with the corresponding explanation in the final version. Thank you for your feedback again.
>
> *Table 1: Comparison with evaluation metrics on CIFAR-10 dataset with malicious proportion* *γ=0.3. We use the trade-off V to evaluate the performance of different evaluation metrics.*
>
> |   | Cosine similarity | Euclidean distance | Wasserstein distance |
> | ---- | :-------------------: | :----------------: | :------------------: |
> | SPMC |**72.98**|45.30|69.09|
>
> **W3: Explanation of hyperparameter λ** **in non-proxy distillation.**
>
> A3: We thank the reviewer for pointing out this issue. Specifically, Equation (9) in the paper is as follows:
> $$
> G_{locgrad} =
> \begin{cases}
> G_d, & \text{if } G_d \cdot G_g \geq 0, \\\\
> G_d - \lambda \cdot \frac{G_d \cdot G_g}{\|G_g\|^2} G_g, & \text{otherwise}.
> \end{cases}
> $$
> where $G_g$ represents the general knowledge between client and its marginal coalition, $G_d$ represents the local knowledge and $G_{locgrad}$ denotes the final local updated gradient.  Next, we analyze the role of hyperparameter λ when $G_d \cdot G_g < 0$. When λ=1, the formula becomes $G_{locgrad} = G_d -\frac{G_d \cdot G_g}{\|G_g\|^2} G_g$.  At this point, $G_{\text{locgrad}}$ is orthogonal to $G_g$, as shown by the following derivation:
>
> $$
> \begin{aligned}
>  G_{locgrad} \cdot G_g &= (G_d - \frac{G_d \cdot G_g}{\|G_g\|_2} G_g) \cdot G_g \\\\
>  &= G_d \cdot G_g - \frac{G_d \cdot G_g}{\|G_g\|_2} \|G_g\|_2^2 \\\\
> &= G_d \cdot G_g - G_d \cdot G_g \\\\
> &= 0.
> \end{aligned}
> $$
>
> When λ>1, we have $G_{locgrad} \cdot Gg>0$. This means the angle between$G_{locgrad}$ and $G_g$ is less than 90°, indicating that their directions are more closely aligned. In this case, the update gradient leans more toward general knowledge, potentially at the cost of missing important local knowledge. As for λ<1, we have $G_{locgrad} \cdot Gg<0$. This means the angle between$G_{locgrad}$ and $G_g$ is more than 90°, indicating that a less noticeable adjustment toward general knowledge, causing the update gradient to stay closer to local malicious knowledge for attacker.  The reviewer might have overlooked **"Figure 3. Comparison of different λ"** in our original manuscript. To further clarify, we present a comparison of different λ values again in **Table 2**. We sincerely appreciate your constructive suggestion, and we will include the corresponding derivation and explanation in the revised version.
>
> *Table 2: Comparison with different lambda on CIFAR-10 dataset with malicious proportion γ={0.2, 0.3}. We use the trade-off V to evaluate the performance of different lambda.*
>
> | $\lambda$ |  1.5  |    1.0    | 0.5  |
> | :-------: | :---: | :-------: | :--: |
> |   γ=0.2   | 60.99 | **85.32** | 2.50 |
> |   γ=0.3   | 51.87 | **80.14** | 3.53 |

---

> > ### Comment · Reviewer_hJgY · 2025-04-03
> >
> > Thank you for the response. The clarifications and added results address my main concerns. Please revise the paper based on your rebuttal. Since the authors addressed my concerns, I decide to raise my score to 4. Good luck!

---

### Decision · Program_Chairs · 2025-05-01

**Decision:**

Accept (poster)

**Comment:**

This paper is novel in its methodology design and provides a novel insight to understanding the malicious backdoor attacks in FL. The presentation is also clear and experiments are sufficient. However, the method requires extra the hyper-parameters, which limits the wide adoption.